# Transgenic East African Highland Banana Plants Are Protected against *Radopholus similis* through Host-Delivered RNAi

**DOI:** 10.3390/ijms241512126

**Published:** 2023-07-28

**Authors:** Henry Shaykins Mwaka, Lander Bauters, Josephine Namaganda, Shirley Marcou, Priver Namanya Bwesigye, Jerome Kubiriba, Guy Smagghe, Wilberforce Kateera Tushemereirwe, Godelieve Gheysen

**Affiliations:** 1Department of Biotechnology, Ghent University, 9000 Ghent, Belgium; henry.mwaka@ugent.be (H.S.M.); lander.bauters@ugent.be (L.B.); 2Department of Plants and Crops, Ghent University, 9000 Ghent, Belgium; shirley.marcou@ugent.be (S.M.);; 3National Agricultural Research Laboratories, Kawanda, Kampala P.O. Box 7065, Uganda; mjnamaganda@gmail.com (J.N.); bwesigyep@gmail.com (P.N.B.); jkubiriba2012@gmail.com (J.K.); tkwilberforce@gmail.com (W.K.T.)

**Keywords:** banana, *Radopholus similis*, nematodes, RNAi, transgenic, pest control

## Abstract

The burrowing nematode *Radopholus similis* is considered a major problem of intensive banana cultivation. It can cause extensive root damage resulting in the toppling disease of banana, which means that plants fall to the ground. Soaking *R. similis* in double-stranded (ds) RNA of the nematode genes *Rps13*, chitin synthase (*Chs-2*), *Unc-87*, *Pat-10* or beta-1,4-endoglucanase (*Eng1a*) suppressed reproduction on carrot discs, from 2.8-fold (*Chs-2*) to 7-fold (*Rps13*). The East African Highland Banana cultivar Nakitembe was then transformed with constructs for expression of dsRNA against the same genes, and for each construct, 30 independent transformants were tested with nematode infection. Four months after transfer from in vitro culture to the greenhouse, the banana plants were transferred to a screenhouse and inoculated with 2000 nematodes per plant, and thirteen weeks later, they were analyzed for several parameters including plant growth, root necrosis and final nematode population. Plants with dsRNA constructs against the nematode genes were on average showing lower nematode multiplication and root damage than the nontransformed controls or the banana plants expressing dsRNA against the nonendogenous gene. In conclusion, RNAi seems to efficiently protect banana against damage caused by *R. similis*, opening perspectives to control this pest.

## 1. Introduction

While banana is an important crop for millions across the world both for staple food and fruit, and is grown in more than 135 countries worldwide, its production has been declining owing to mounting pressure from pests and diseases. The major diseases include Fusarium wilt [1], Black sigatoka [2], Banana bacterial wilt [3] and banana streak virus [4]. The major pests include Banana weevil [5] and nematodes [6]. While up to 34 plant parasitic nematodes have been reported to infest banana [7], the economically important nematodes include the burrowing nematode (*Radopholus similis*) [8,9], root-knot nematodes (*Meloidogyne* spp.) [10], lesion nematodes (*Pratylenchus* spp.) [11] and spiral nematodes (*Helicotylenchus* spp.) [12]. When a comparison of associated root damage was made between *Pratylenchus goodeyi*, *H. multicinctus* and *R. similis*, the latter showed stronger correlation with root damage [13]. Indeed, *R. similis* is considered the main problem of intensive banana cultivation [14,15] and has often been referred to as the toppling disease of banana [16,17,18], where plants with extensive root damage fall to the ground [19,20,21]. The burrowing nematode, which was first identified in banana plantations in Fiji [8], primarily exists in the tropical and subtropical climates, at low elevations where the temperature ranges between 24 and 32 °C [14]. *R. similis* was spread through the distribution of infected planting materials [22,23,24]. Once introduced in a new field, infestation begins when *R. similis* intrudes into the roots and gains entry into the cortex, where an adult female lays up to six eggs per day for a few weeks [25,26]. The nematode completes its life cycle in the root within three weeks at 25–30 °C [27], though adaptation to lower temperatures has been reported [25]. The distinct symptoms of infestation can be visualized when longitudinally dissected roots or peeled corms display reddish-brown lesions [28] which degenerate into black rotting tissue, resulting from proliferation of microbes [29,30]. Other general symptoms include low vigor, stunted growth, yellowing, drooping leaves and reduced yield [29]. *R. similis* has been reported to cause damage ranging between 30 and 80% [15,31], with yield losses of up to 12.5 tons/ha [32], which could explain the low yields in many subsistence farming systems in sub-Saharan Africa [33,34]. The damage to the root system impairs uptake of water and nutrients to support plant growth [35], making the plants very susceptible to drought. The extent of the damage caused by nematodes has often been underappreciated, because general symptoms are frequently misdiagnosed as nutrient deficiency, drought or unknown causes [36]. The successful spread of *R. similis* has been attributed to its wide host range of more than 365 plant species [37], short life cycle and hermaphroditism [25,38,39]. The importance of controlling banana nematodes was underscored when the use of nematicides resulted in a 275% increase in yield [40], which makes the finding important in the context of a rapidly growing population and ever-reducing land available for agriculture [41,42]. As an endoparasitic migratory nematode, *R. similis* completes its life cycle in the root [43], making it difficult to diagnose and control. Control of banana nematodes is also difficult, because cultural interventions are not easily applicable owing to the perennial nature of the crop. Fumigation, an expensive alternative to control nematodes, presents health and safety concerns on the environment, which renders it unsuitable [44]. The most sustainable way to manage nematodes will therefore depend on the availability of resistant farmer-preferred varieties. Breakthroughs in plant biotechnology provide an important avenue for mitigating declining yields attributed to nematodes. Over the last 20 years, there has been tremendous success in generating transgenic bananas using embryogenic cell suspensions [45]. Genetic engineering has been exploited to provide resistance against nematodes in banana using protease inhibitor genes [46,47]. While traditional transgenic approaches could be argued to potentially have some risk on nontargets [48], RNAi technology based on host-delivered dsRNA could potentially address several biosafety concerns, perceived, imagined or otherwise, because no new protein is produced in the transgenic plants [49]. RNAi is a naturally occurring conserved process responsible for gene regulation and defense against pathogens [50,51]. The use of RNAi in pest control is arguably safe owing to new insights from studies demonstrating that dsRNA in soils and plant debris is prone to rapid degradation [52,53], allaying fears of environmental persistence. Furthermore, since living organisms have evolved to break down dsRNA and use the nucleic acids as cellular nutrients, this technology will be safer than conventional chemistries for those who apply RNAi products or eat the product [54]. In deploying RNAi for use in the control of pests, it is imperative that target genes are carefully selected and exhaustively validated considering that response to RNAi may be variable [55]. Since the efficacy of RNAi varies depending on factors such as target gene selection, method of dsRNA delivery and expression of dsRNAs, this study focused on the genes *Chs-2*, *Ego-1*, *Eng1a*, *Pat-10*, *Rps13* and *Unc-87*, previously demonstrated to be effective targets for RNAi in nematodes. Chitin synthase (*Chs-2*) was selected, because viable egg production requires chitin, which is a key component of the eggshell [56]. Ingestion of dsRNA by *C. elegans* and soaking eggs of *Meloidogyne artiellia* in *Chs-2* dsRNA affected egg development [57]. Host-induced gene silencing of chitin synthase conferred resistance to soybean cyst nematode in soybean [58]. The *Ego-1* gene encodes an RNA-directed RNA polymerase that is important in *C. elegans* germline development [59]. Endo-β-1,4-glucanase (*Eng1a*) hydrolyzes cellulose, allowing nematodes to degrade and penetrate cell walls [60], and RNAi has resulted in reduced nematode infection [61]. The *Unc-87* gene encodes actin-binding proteins essential for maintenance of the nematode body wall muscle, which plays a critical role in worm motility [62,63]. Related to movement is *Pat-10* encoding a body-wall muscle troponin C, essential for muscle contraction and successful embryonic morphogenesis [64]. RNAi of *Pat-10* has been shown to result in paralysis (“walking stick” phenotype), larval and embryonic lethality and maternal sterility in the nematode [65,66]. The *Rps13* gene encodes the ribosomal protein S13, a component of the 40S subunit of ribosomes that catalyze protein synthesis [67]. *Rps13* has been reported to be important in germline homeostasis, and RNAi resulted in reduced fecundity [68].

## 2. Results

### 2.1. Effect of dsRNA on Multiplication of Radopholus similis on Carrot Discs

To evaluate the effect of double-stranded RNA (dsRNA) on multiplication of *R. similis*, first an in vitro assay was set up using carrot discs. To compare the efficacy of the RNAi for the different target genes, a mixed-stage sample of *R. similis* was incubated with dsRNA, and, after soaking, the nematodes were inoculated on mini carrot discs. Eight weeks later, the carrot discs were inspected, after which the nematodes were extracted and counted. A visual inspection of the carrot discs from each of the treatment batches indicated differences in the level of degradation. The most degraded carrot discs were the negative controls (dsCs-Laccase-2 and no dsRNA treatments). The least degraded carrot discs were from the dsRps13 treatments.

A one-way ANOVA was conducted to compare the effect of dsRNA on multiplication of *R. similis* on carrot discs after soaking in dsRNA of nonendogenous and target genes and no dsRNA conditions. There was a significant effect (*p* < 0.001) for the various dsRNA conditions. The highest nematode counts were observed in the dsCs-Laccase-2 and no-dsRNA negative controls. Post hoc comparisons using the Tukey HSD test indicated that all the other dsRNA treatments registered significantly lower nematode counts (summarized in Table 1). The dsRps13 treatment had the lowest nematode count with an 8-fold reduction in nematode multiplication, followed by dsUnc-87, dsEng1a, dsEgo-1, dsPat-10, all about a 4–5-fold difference, and finally dsChs-2 (2.8-fold reduction).

### 2.2. RT-PCR Analysis of the Soaked Nematodes

A multiplex RT-PCR to compare expression levels was set up each with two sets of primers, one amplifying a region within the actin gene and the other a region within the specific target genes (*Ego-1*, *Rps13*, *Eng1a*, *Chs-2*, *Pat-10* and *Unc-87*). Some of the PCR results are shown in Appendix A. The relative expression of the various genes as computed using ImageJ showed a range of 3.8-111-fold reduction in expression levels of treatment compared to control (Figure 1).

### 2.3. Generation of Transgenic Banana with Hairpin Constructs to R. similis Genes

After cocultivation of embryogenic cell suspensions with *Agrobacterium* harbouring the RNAi hairpin cassette, the cells which had been thinly spread out on MA3 medium supplemented with kanamycin and cefotaxime gradually regenerated into embryos and plantlets (Figure 2). Within a two-month period, both cell death (black tissue) and growth (white embryos) were observed. The embryos were placed on embryo development media and eventually transferred to proliferation and rooting media.

Plantlets were left in culture for about four weeks to develop sufficient leaf tissue for molecular analysis prior to multiplication of lines for screenhouse evaluation.

Once the plantlets developed at least one gram of tissue, they were subcultured, during which leaf tissue was cut for molecular analysis. All plantlets were first screened for the *nptII* gene to confirm transformation. A gene-specific PCR was then used to check the presence of the gene-specific hairpin construct. A primer that binds to a region of the intron and another primer from each gene was used to confirm the successful transformation events (Figure 3). Screening for transformants was stopped once 50 lines per construct were obtained.

The confirmed transgenic plants were proliferated (five clones per line), rooted, transferred to the greenhouse and placed in a weaning chamber for three weeks to acclimatize. After weaning, the plants were grown for three months prior to infection with nematodes.

### 2.4. Necrosis and Nematode Multiplication in dsRNA Plants

Three out of the five replicates from each transgenic line were prepared for infection tests, as described in Section 4.6.4. A total of 90 plants per construct (30 lines, 3 replicates each) were infected with a mixed-stage population of nematodes. Data were collected from each of the 90 plants per construct comprising 30 different transgenic lines replicated thrice and set up in a completely randomized design. The plants were assessed for root damage three months after inoculation. When the roots were washed, it was evident that every inoculated plant had experienced some level of *R. similis* infestation with varying degrees of success. It was easy to identify plants with high infestation, which showed a comparatively higher number of dead roots, in contrast to those with fewer dead roots (Figure 4).

In addition to the general observed differences in infestation, lesions due to nematode migration were visible. Initial damage was observed as small dark purplish-red lesions on the outer part of the roots and cortical tissue. On dissection, it was observed that there were intense differences in discoloration of the cortex tissue among the test plants (Figure 5), with generally more uniform discoloration in the control (wild-type) plant roots extending throughout the cortex.

A Kruskal–Wallis test followed by a Dunn’s test revealed the nonendogenous control (dsLaccase-2) and NTC to have a significantly higher necrosis index compared to the other five constructs, but they do not differ significantly from each other. There are also some significant differences between other pairs of groups. Specifically, within the five other transgenic plant groups (dsRps13, dsUnc-87, dsChs-2, dsEng1a, dsPat-10), the dsRps13 and dsChs-2 plants appear to have a significantly lower necrosis index compared to the dsEng1a and dsUnc-87 constructs (Figure 6). Appendix A gives an overview of all transgenic and control lines tested, and the best performing lines showing the lowest necrosis are listed in Table 2.

Apart from the necrosis index as a measure for resistance, parameters on vegetative growth were analyzed to determine whether better performance was synonymous with lower infestation counts. The parameters considered were number of live and dead roots, plant height and fresh root weight. There was a strong relationship between plant height (Appendix A) and a lower necrosis index for dsRNA lines compared to the noninoculated controls. Simple linear regression was used to test whether plant height significantly predicted necrosis index. The overall regression was statistically significant (R^2^ = 0.87, F(1, 628) = 3770, *p* < 0.001). It was found that plant height significantly predicted necrosis index (β = −0.614* necrosis index, *p* < 0.001), and plant height predictions for the various test plant groups are provided in the regression equations (Appendix A).

To determine whether the female nematode populations (Rs-fem in Appendix A) were different in the roots of the seven plant groups, a Kruskal–Wallis test was carried out followed by a Dunn’s test. This showed that, while a significantly higher number of females were counted in the nonendogenous control (dsLaccase-2) and the nontransformed control, they do not differ significantly from each other. However, they are significantly different from the five target gene constructs, as visualized in the box plot (Figure 7). Similarly to the results on necrosis index, dsRps13 and dsChs-2 had a significantly higher number of females in comparison with the dsUnc-87 and dsEng1a plant groups. On the other hand, dsPat-10 plants do not seem to differ statistically from the four other target gene constructs (dsRps13, dsUnc-87, dsChs-2, dsEng1a).

In conclusion, the controls had higher median values than the test plants, indicating that they provided a better environment for multiplication of the nematodes. While in the nontransformed and the nontarget controls, only 6 replicates of 180 (scattered over six different lines) multiplied no nematodes, some of the transgenic lines had three replicates with no nematodes (such as Nak-Eng-53 and Nak-Pat-10-104, Appendix A).

## 3. Discussion

In the past decade, there has been considerable research on the use of RNAi to protect crops from pests and pathogens [69]. RNAi, which targets specific transcripts within a pest species and has no negative effects on nontarget species, has the potential to create a new generation of targeted pesticides because of its sequence specificity [70]. Early investigations in insects and the nematode *Caenorhabditis elegans* typically supplied double-stranded RNA (dsRNA) through ingestion [71], injection [72], and feeding via microorganisms producing dsRNAs [73,74].

Urwin et al. (2002) were the first to demonstrate that soaking plant parasitic nematodes with in vitro-generated dsRNA was effective in reducing the mRNA levels of the targeted genes. However, this strategy cannot be used to manage plant parasitic nematodes in agricultural settings. Host-delivered dsRNA is an efficient way to continuously supply dsRNA to parasitic nematodes in (crop) plants. It was first shown to decrease rootknot nematode infection in *Arabidopsis thaliana* [75] and has since been demonstrated to work against different plant parasitic nematodes.

After various in vitro assays confirming the susceptibility of *Radopholus similis* to posttranscriptional gene silencing, in planta RNAi was eventually validated for protection against *R. similis* by targeting the cysteine proteinases cathepsin B [76] and cathepsin C [77], resulting in tomato plants with drastic decreases in nematode reproduction. In addition, tomato plants expressing dsRNA to *R. similis* calreticulin were reported to show resistance to infection by the nematode [78]. In banana, successful RNAi against a pest was validated when transgenic plants expressing aphid *AChE* dsRNA resulted in up to 75% protection [79]. In our study, we report the first attempt of applying host-delivered dsRNA against *R. similis* using transgenic banana. To expand the RNAi toolbox, we focused on different genes and based our criteria by taking into consideration confirmed RNAi success in (other) nematodes, potential intellectual property considerations, regulatory compliance, target specificity and off-target effects. The following genes were selected as a target for this study: *Chs-2*, *Ego-1*, *Eng1a*, *Pat-10*, *Rps13* and *Unc-87.*

In this study, we explored the potential of dsRNA against these selected target genes to control *R. similis* by initially evaluating the effect of dsRNA soaking of the nematodes on their multiplication on carrot discs. Carrots discs provide an ideal method for evaluating multiplication of nematodes, because they are living plant tissue, similar to the environment in the banana root. As obligate parasites, upon entering the carrot tissue, *R. similis* nematodes feed on the cytoplasm of cortex cells, as such destroying them and causing cavities to develop. The greater the numbers, the greater the damage. In optimal conditions, the *R. similis* female produces an average of about 2 eggs/day (range is 0.5–6 eggs/day/female), which can hatch in 3–7 days and complete a life cycle in 18–20 days, which implies that the 55 days is sufficient for at least 2 generations. The starting number of 30 adults, after a period of 55 days, were capable of multiplying to several thousand nematodes depending on the state of the carrot disc. As the results suggest, the dsRNA treatments significantly affected the multiplication of the nematodes on the carrot discs. The highest numbers of *R. similis* (>18,000) recovered were from the negative controls (no dsRNA and dsCs-lac). The multiplication rates were consistent with observed differences in the relative expression of target genes following dsRNA soaking. For instance, in comparison to the nonendogenous dsRNA control, there was a 91-fold reduction in *Rps13* expression, which corresponded to an 8-fold reduction in nematode counts on carrot discs.

The consequences of soaking nematodes in dsRNA of the selected target genes on their multiplication in carrot discs encouraged further investigation into host-delivered dsRNA using transgenic banana plants. Thus, the follow-up study generated and evaluated the performance of transgenic banana with dsRNA constructs against *R. similis*. The findings show significant differences attributed to the potency of the different dsRNA, with dsRps13 and dsChs-2 plants showing the highest protection against *R. similis*-caused damage and the greatest suppression of nematode multiplication. Out of the 30 dsRps13 transgenic lines tested, there were significant differences in necrosis index levels, nematode counts, dead roots, live roots and plant height. For instance, lines 86 and 127 exhibited very high protection in contrast to lines 123 and 43, which showed high susceptibility to root damage associated with high nematode reproduction (Appendix A). The difference in protection could be attributed to various factors such as the transgene copy number, position effects and transcriptional interference.

Comparing the multiplication rate of nematodes on carrot discs (Table 1) with the number of *R. similis* females recovered from the roots of transgenic plants (Figure 7) showed interesting similarities and differences between dsRNA treatments. While dsChs-2 was the least effective in reducing nematode multiplication on carrot discs in the soaking experiment, it was among the two most effective dsRNA in the transgenic banana plants. The difference could perhaps be attributed to the role and expression pattern of *Chs-2*. Chitin synthase is mainly expressed in adult females and is important for the reproduction of the nematode, as chitin is a crucial component of the eggshell. When mixed-stage nematodes were soaked in dsChs-2, the silencing effect on adult females was reflected in a reduced *Chs-2* mRNA level, but the apparent limited effect on their multiplication could be due to the transient nature of the silencing after soaking [80]. Once the nematodes are placed on carrot discs with no further supply of dsRNA, they are able to partially recover. As the nematode inoculum comprised different developmental stages, it is possible that the nematodes soaked as juveniles are not much affected anymore by the dsRNA once they develop into egg-laying females. In contrast, the other target genes are expressed in, and are important for, the functioning of all nematode stages, so any negative change in their expression levels will result in an immediate negative effect on the survival of all stage nematodes. In the case of transgenic plants producing chitin synthase dsRNA, the RNAi effect is sustained and can persist through all stages and subsequent generations. As a result, all nematode females in the transgenic plants are exposed to a steady dose of dsRNA and will be affected in their egg production, resulting in a much lower number of nematodes recovered from the roots of the transgenic plants expressing dsChs-2.

The genes that were used in this study were carefully selected, taking into account specificity to nematodes. Overall, RNAi technology offers a safe and effective alternative to traditional pest control methods, with minimal impact on nontarget organisms and the environment. The safety arises from target specificity of selected genes used for protection against nematodes. Not only were genes selected for control of nematodes unique to plant parasitic nematodes, they are also unique to *R. similis*, with sufficient sequence diversity to drastically reduce any chance of off-target effects in other organisms. The mean G + C content of the nucleotides at the third codon position (GC3%) in *R. similis* was reported to be as high as 64.8%, the highest for nematodes reported to date [81]. Therefore, the corresponding genes in other parasitic nematodes of banana (such as *Pratylenchus*) are not similar enough in sequence to be targeted by the same dsRNAs. While it may be desirable to have a single pesticide that is effective against a wide range of pests or diseases, the specificity of RNAi-based pesticides is one of their key strengths. In fact, RNAi-based pesticides can be designed to specifically target the genes of a particular pest or pathogen, while leaving nontarget organisms unharmed. This is in contrast to traditional chemical pesticides, which can have broad-spectrum effects that harm beneficial organisms and disrupt ecosystems. The use of RNAi provides a wide safety margin right from the stage of design, since prospective target sequences can be evaluated using publicly available bioinformatic tools and readily available gene sequence databases.

In conclusion, this study focused on the potential of host-delivered dsRNA against the nematode *R. similis*, which poses a significant threat to banana crops. The results showed that transgenic banana plants expressing dsRNA constructs provided significant protection against *R. similis*. While this study was limited to potted plants in a screenhouse for a limited period of time, field evaluation of the best transgenic banana plants expressing dsRNA constructs against nematodes should be done. By reducing the damage caused by nematode infestations, the plants may be able to allocate more resources towards growth and development, leading to higher yields. Nevertheless, field evaluations would be different from the potted trial conducted in a screenhouse, because field conditions are more complex and variable, and can include factors such as climate, soil type, and other pests and diseases.

The use of RNAi for pest and pathogen control in agriculture shows great promise, and future perspectives are quite exciting. Finally, there is a need for greater collaboration between researchers, regulatory bodies, and industry stakeholders to ensure that the development and deployment of RNAi-based pest and pathogen control methods are safe, effective, applicable and sustainable.

## 4. Materials and Methods

### 4.1. Nematode Population

The nematodes were obtained from roots of symptomatic Mbwazirume banana plants hosted at the National Agricultural Research Laboratories in Kawanda, 13 km North of Kampala (0°25′ N, 32°32′ E) and at an elevation of 1195 m above sea level. The average daily temperatures range between 15 °C and 29 °C, with a mean relative humidity of 76%. The locality receives mean annual rainfall of about 119 mm per year in a bimodal distribution.

*R. similis* from infested roots was extracted through maceration and filtration as previously described [82,83]. Briefly, 5 root pieces assessed for cortical necrosis were cut into 1 cm long segments and mixed. A 5 g subsample was then added to 50 mL of water and blended at medium speed for 10 s. The macerate was poured onto two 2-ply paper towels (Bounty, Procter and Gamble, Cincinnati, OH, USA) suspended on a Baermann tray with the water level in the tray touching the roots [84]. Two days later, the water suspension with nematodes was removed, replaced with fresh water, and the nematodes counted in two 2.5 mL aliquots of the suspension. To obtain a pure culture of *R. similis*, adult females were identified using a dissecting microscope and picked out using distinguishing features such as the long stylet, elongated tail and vulva, as previously described [85].

### 4.2. Culture of R. similis on Carrot Discs and Extraction of Nematodes

A nematode culture of *R. similis* was established under axenic conditions as previously described [86]. Single *R. similis* adult females were placed on a droplet of sterile water at the center of previously prepared 2-week-old carrot discs, sealed with cling film and maintained in the dark at 25 °C. After transport to Belgium, the nematodes were maintained in a 28 °C growth room at Ghent University with bimonthly subcultures on fresh carrot discs.

At 7 weeks, the plates which showed nematodes actively swimming in a water film surrounding the carrot discs were selected for extraction. The petri dishes were rinsed to collect the nematodes while the carrot discs were soaked in sterile water for 1 h, stirred and sieved into a container. The carrot discs were cut into pieces and soaked again for 30 min to recover the remaining nematodes. The nematodes were placed into a 500 µL volume by filtration and decantation. The number of nematodes was determined by taking aliquots (10 µL) and counting the nematodes in six technical replicates under a dissecting microscope at 40× magnification using a gridded counting dish and tally counter. The average number of nematodes extracted was computed based on these counts.

### 4.3. Target Gene Selection

*R. similis* orthologs of the selected targets were identified with TBLASTN using known protein sequences of plant parasitic nematodes against *R. similis* transcriptome data (shared by Blaxter Lab University of Edinburgh, Edinburgh, UK). Target regions from selected genes were chosen after analysis to identify sections with a higher probability of generating potent small interfering RNAs (siRNAs). Briefly, selected gene sequences were analyzed with software from the Whitehead siRNA Selection Web Server [87] to assess a region with the highest potential for generating potent siRNAs and identify sections with no off-target hits using the human and mouse genomes. To assess potential off-target effects of the selected RNAi sequence, we employed the NCBI nr/nt database, a comprehensive resource for nonredundant nucleotide sequences. The search parameters were set to search against the nr/nt database, ensuring comprehensive coverage of available nucleotide sequences. Upon completion of the BLAST search, we carefully analyzed the results to identify any potential off-target matches or similarities. Factors such as E-values and alignment scores were considered, with lower values indicating higher sequence similarity. Matches with sequences from unintended organisms or genes that could lead to off-target effects were particularly scrutinized. To further evaluate the conservation and potential functional homologs of the identified off-target hits, we performed additional comparisons against Insecta nr/nt databases. This step allowed us to assess the likelihood of off-target effects and identify RNAi sequences with high specificity and limited potential for unintended consequences. In summary, the utilization of the NCBI nr/nt database, in combination with careful analysis and evaluation, served as a critical step in our target selection process. It facilitated the identification and mitigation of potential off-target effects, reinforcing the reliability and specificity of our RNAi-based experimental design. The genes were amplified, cloned and confirmed by sequencing, as described below. The target sequences are shown in Appendix A.

### 4.4. RNA Isolation, cDNA Synthesis and Cloning Targets

#### 4.4.1. RNA Isolation and cDNA Synthesis

RNA was extracted from several batches of approximately 1000 mixed-stage nematodes using sonication [88] and phase separation with Tri-reagent (Sigma-Aldrich, Saint Louis, MO, USA) and chloroform [89]. Trace amounts of genomic DNA were removed by incubating for 30 min at 37 °C with RNase-Free DNase in an 18 µL reaction comprising 1.8 µL of buffer with Mg, 1 µL of RNAse inhibitor (Thermo Fisher Scientific, Merelbeke, Belgium), 1 µL of DNAse (DNAse I, Thermo Fisher Scientific, Merelbeke, Belgium) 10 µL of RNA and 4.2 µL of RNAse-free water.

RNA was reverse-transcribed with oligomer (dT30) as a primer using the SuperScript III First Strand Synthesis System for RT-PCR (Invitrogen, Carlsbad, CA, USA) by following the manufacturer’s instructions. The 40 µL setup, which comprised 20 µL of DNAse-treated RNA, 1 µL of oligo dT (700 ng/µL), 8 µL of 5× first-strand buffer, 4 µL of 0.1 M DTT, 2 µL of 10 mM dNTPs, 1 µL of Superscript Reverse Transcriptase and 4 µL of RNAse-free water was incubated for 2 h at 42 °C. The cDNA reaction products were stored at −80 °C with aliquots stored at −20 °C.

#### 4.4.2. Generation of Vectors for the Production of dsRNA

Selected regions of the different target gene were amplified from cDNA by PCR using a Bio-Rad Thermocycler (Bio-Rad, Hercules, CA, USA) with the primers and annealing temperatures as listed in Appendix A. Other PCR conditions were an initial denaturation step at 94 °C for 5 min, followed by 35 cycles of 94 °C for 30 s, 30 s allowed for annealing, an extension at 72 °C for 45 s, finishing with an extension step at 72 °C for 10 min. The PCR products obtained were separated by gel electrophoresis and bands with expected sizes excised and purified, as previously described [90]. The purified PCR products were cloned in pGEM^®^-T-Easy vector (Promega Benelux, Leiden, The Netherlands,) using TA cloning [91] and selected using a combination of blue-white screening [92] and colony PCR [93].

To produce dsRNA, the inserts from selected pGEMT clones were excised and subcloned [94,95] in the MCS located between the bidirectional T7 promoter of the Litmus38i vector. Restriction enzymes *Apa*I and *Sal*I (New England Biolabs, Ipswich, MA, USA) were used to excise the insert from the recombinant PGEMT donor and linearize the Litmus 38i destination vector according to manufacturer’s instructions. To improve efficiency of ligation, the resulting products were processed through dephosphorylation and separated by gel electrophoresis [96]. Electrophoresis products were purified using the standard protocol of the Silica bead DNA Gel extraction kit (# K0513, Thermo Fisher Scientific, Merelbeke, Belgium). The constructs were then transformed into the *E. coli* strain HT115 (DE3), which is deficient in RNase III [74,97], using heat shock [98]. Transformed colonies were screened by colony PCR [93,99].

#### 4.4.3. Generation of Vectors for Plant Transformation

To assess RNAi-induced protection conferred by host-delivered dsRNA against *R. similis*, six hpRNA constructs were constructed in pCambia2300: *Chs-2* (477nt), *Eng1a* (500nt), *Pat-10* (420nt), *Rps13* (388nt), *Unc-87* (393nt) and *Ego-1* (312nt). pCambia2300-cs-Laccase-2 was used as a negative control (nontarget). Each vector harbored a sense and antisense target sequence linked through an intron and under control of the *Zea mays* Ubi promoter. A target sequence of a few hundred base pairs is known to be functional for gene silencing [100]. Therefore, the 300–500 bp sequences of the different genes were cloned in the vector pCambia2300.

For subcloning into the hairpin RNAi vector construct, sequenced PGEMT-Easy vector clones with confirmed inserts in functional orientation, from coding sequences of the selected genes (see Section 4.4.2), were used as a starting point to generate PCR products for vector construction using sticky-end PCR [101]. In the initial step, primers flanking sections of target genes were designed and incorporated with 4 different restriction sites (*Not*I, *Xho*I, *Mlu*I and *Bam*HI) for directional cloning in the intermediate cloning vector, the Banana Master Construct (BMC). *Not*I and *Xho*I (New England Biolabs, Ipswich, MA, USA) were incorporated in the 5′ end of the forward and reverse primers to amplify a PCR product for directional cloning in sense orientation while, similarly, *Bam*HI and *Mlu*I (New England Biolabs, Ipswich, MA, USA) primers directed antisense orientation. Sequenced PGEMT-Easy clones were used as templates to amplify targets. In the next step, PCR was performed using a Bio-Rad Thermocycler with the primers and annealing temperatures listed in Appendix A. Thirdly, PCR products obtained were fractionated by gel electrophoresis and bands with expected sizes excised and purified as previously described [90]. For sticky-end ligation and cloning in the hairpin construct, the cleaned PCR product was double-digested with either *Not*I-HF^®^ and *Xho*I or *Bam*HI-HF^®^ and *Mlu*I-HF^®^ (New England Biolabs, Ipswich, MA, USA), depending on the primer combination used. Insertion of sense and antisense products in the destination vector was done sequentially. The BMC vector was initially linearized through a *Not*I-HF^®^ and *Xho*I digest, cleaned up and ligated to cleaned-up PCR product with compatible sticky ends in a sense orientation of target gene fragments, followed by cloning and selection of recombinant colonies with the correct sizes. Similarly, antisense fragments were inserted using *Bam*HI-HF^®^ and *Mlu*I-HF^®^ digestion and ligation of the corresponding cleaned PCR product in antisense orientation.

For the plant transformation binary vector, standard cloning techniques were used to insert approximately 3.8 kb of the dsRNA-forming hairpin sequence from the various recombinant BMC donors into the 8.7 kb pCambia 2300 binary vector to create the final plant transformation vectors. Both the donor and recipient were digested with *Eco*RI-HF^®^ and *Hin*dIII (New England Biolabs, Ipswich, MA, USA) in a 50 µL reaction mixture according to manufacturer’s instructions. After electrophoresis on a 1% TAE agarose gel and cutting out the appropriate fragment, the hairpin sequence was purified by resin binding [102] using the Silica bead DNA Gel extraction kit (# K0513, Thermo scientific). The linearized pCambia2300 (backbone) was purified from the *Eco*RI-HF^®^ and *Hin*dIII double-digest reaction mix through ethanol precipitation with sodium acetate [103]. Ligation reactions of insert and backbone were set up as described by [104], using a vector-to-insert ratio of 1:3, as recommended [105]. The ligation mix was transformed into *E. coli* JM109 [98] followed by a miniprep and diagnostic digests with *Eco*RI-HF^®^ and *Hin*dIII to select recombinant clones.

### 4.5. RNAi of R. similis by Soaking in dsRNA

#### 4.5.1. Synthesis and Isolation of dsRNA from Bacteria

For production of dsRNA, the *E. coli* HT115 (DE3) strain was used with the recombinant Litmus38i plasmids (Section 4.4.2), and then the produced dsRNA was purified as previously described [106,107]. The *E. coli* HT115 containing the recombinant Litmus38i with selected target sequences were cultured in 4 mL of LB (10 g/L tryptone, 10 g/L o yeast extract, 5 g/L of NaCl at pH 7.0) broth medium supplemented with carbenicillin (100 μg/mL) and tetracycline (12.5 µg/mL) at 37 °C, maintained at 200 rpm overnight on a rotary incubator. The following day, a 250 µL aliquot of overnight culture was used to inoculate 25 mL of LB supplemented with the same antibiotics, which was incubated on a 200 rpm rotary incubator at 37 °C until an OD_600nm_ of 0.4 was attained when IPTG was added to a 1 mM concentration to activate the T7 promoter for RNA transcription. The cells were harvested by centrifugation at 7000 rpm for 10 min and resuspended in 8 mL of physiological solution. Next, 1 mL aliquots were centrifuged at 7000 rpm for 10 min in 1.5 mL tubes and stored at −80 °C. Extraction and purification of dsRNA from the bacterial cells involved cell lysis with TES buffer (10 mM Tris pH 7.5, 10 mM EDTA and 0.5% SDS), ssRNA was removed by incubation with 5 µL of RNase A (1000 U/µL; Thermo Fisher Scientific, Merelbeke, Belgium) in 10× RNase A buffer (4 M NaCl, 0.1 M Tris-HCl), while dsRNA was purified using TRI reagent (Sigma-Aldrich, Saint Louis, MO, USA) followed by extractions with chloroform and ethanol as previously described [108]. To quantify dsRNA obtained, the actual absorbance was determined with a Nanodrop 2000 UV-visible spectrophotometer (Thermo Fisher Scientific, Waltham, MA, USA), and concentration was estimated using a conversion factor of 46.74, as previously described [109].

#### 4.5.2. Preparation of Mini Carrot Discs and Soaking Solution

Carrots with green tops obtained from Delhaize Supermarket (Gent, Belgium) were surface-sterilized (Moddy et al., 1973) and diced into 15 mm thick slices. A 12 mm diameter cork borer was used to cut out mini discs which were singly placed in wells of sterile 24-well plates (Greiner Bio-One, Frickenhausen, Germany), which were covered with a sterile lid, sealed with parafilm and incubated at 25 °C for 2 weeks.

A 100 mL volume of 5X M9 soaking buffer was prepared from 43 mM Na_2_HPO_4_, 22 mM KH_2_PO_4_, 2 mM NaCl, 4.6 mM NH_4_Cl), pH was adjusted to 7.5 using potassium hydroxide and the media was supplemented by gelatin and spermidine, as shown in Appendix A.

#### 4.5.3. Soaking Procedure

The nematodes were extracted from the carrot discs as described in Section 4.2. A 14.38 µL nematode suspension containing approximately 70 nematodes per µL was added to a tube containing a soaking solution comprised of dsRNA (final concentration of 2 mg/mL), 3 mM spermidine (S-2626, Sigma-Aldrich, Saint Louis, MO, USA), 0.05% gelatin in a final volume of 30 µL.

The soaking tubes were made in six biological replicates for each dsRNA (dsChs-2, dsEgo-1, dsEng1a, dsRps13, dsPat-10 and dsUnc-87), and they were incubated at 25 °C for 36 h on a rotary shaker set at 50 rpm. Similarly, a control treatment included nematodes suspended in the soaking solution without dsRNA and nematodes suspended in soaking buffer supplemented with a nonendogenous dsRNA targeting Banana weevil *Laccase-2* [107].

After incubation, the nematodes were rinsed with sterile tap water, centrifuged (1200 rpm for 2 min) and resuspended in sterile tap water. This procedure was repeated twice to completely remove traces of the soaking solution.

#### 4.5.4. Analysis of *R. similis* Reproduction on Mini Carrot Discs

The 24-well plates with carrot discs, free from contamination following the onset of callus, were inoculated with 30 living mixed-stage *R. similis* nematodes previously soaked in dsRNA for a period of 36 h. Twenty-four replicates (discs) per gene were used, they were maintained in the dark for 55 days and nematodes were extracted on the 56th day.

The nematodes were extracted as previously described [110]. The mini carrot discs were cut into small pieces and poured onto the center of a modified extraction plate lined with tissue paper in a sieve. Water was carefully added under the sieve onto the collection plate so as to soak up the tissue and carrot slices. The extraction system was allowed to stand for 24 h. The sieve was removed from the collection plate and the filtrate in the collection plates was transferred into labelled collection tubes. The large volume of liquid was decanted and reconstituted with distilled water thrice, and the solution containing the remaining nematodes was eventually reconstituted in 5 mL. To estimate the nematode population, 1 mL aliquots were drawn from which 6 technical replicates comprising 10 µL aliquots were set out on a petri dish to determine the number of observed nematodes through counts. The total number of nematodes observed from each aliquot was recorded and an average count computed. After a test for normality and one for homogeneity of variance, one-way ANOVA between treatments was conducted to compare the effect of the dsRNA soaking treatments on the multiplication of nematodes over a period of 55 days. Post hoc comparisons were done using the Tukey HSD test.

#### 4.5.5. RT-PCR for Expression Analysis after Soaking

RNA was extracted and cDNA was prepared, as described in Section 4.4.1, from the six replicated batches of nematodes that had been soaked with dsRNA. The cDNA samples were combined per two to have three replicates for the RT-PCR, except for dspat-10 (only two replicates).

A multiplex PCR with gene-specific primers for *Unc-87*, *Pat-10*, *Rps13*, *Chs-2*, *Ego-1* and *Eng1a* was set up each alongside a housekeeping gene, actin in a 25 µL reaction. PCR conditions comprised an initial denaturation at 95 °C for 3 min, with various annealing temperatures (Appendix A) for 30 s, an extension step at 72 °C for 30 s repeated for 27 cycles and a final extension step at 72 °C for 5 min. The PCR products were subjected to electrophoresis on a 1% agarose gel, with images analyzed by ImageJ. Gene expression was normalized between samples by using the internal actin control.

### 4.6. Generation and Testing of Transgenic Banana Plants with RNAi against R. similis

#### 4.6.1. Generation of the Agrobacterium Strains and Culture

The plant transformation constructs were generated as described in Section 4.4.3. Following verification through PCR and restriction analysis, 7 pCambia 2300 RNAi hp constructs were multiplied in the *E. coli* strain JM109 using LB broth supplemented with kanamycin 50 µg/mL, rifampicin 25 µg/mL, carbenicillin 250 µg/mL and incubated at 37 °C overnight on a rotary incubator set at 200 rpm. DNA was extracted using a QIAGEN QIAprep Spin Miniprep Kit (QIAGEN, Venlo, The Netherlands) [111]. Then, 1 µg of pCambia2300 RNAi constructs was introduced into 100 µL aliquots of competent cells of *Agrobacterium* strain AGL1 using the freeze–thaw method [112,113]. The mix was plated on solid Yeast–Mannitol (YM) selective medium (10 g/L of mannitol, 0.4 g/L of yeast extract, 0.1 g/L of K_2_HPO_4_, 0.4 g/L of KH_2_PO_4_, 0.1 g/L of NaCl, 0.2 g/L of MgSO_4_·7H_2_O, pH 6.8) [114] supplemented with kanamycin 50 µg/mL, rifampicin 25 µg/mL, carbenicillin 250 µg/mL and incubated at 28 °C for 2 days. Distinct recombinant colonies were picked, streaked on YM agar plates supplemented with antibiotics, and grown for 3 days at 28 °C. The presence and integrity of the constructs were checked by a combination of colony PCR and analysis of plasmid which was extracted from *Agrobacterium* using alkaline lysis, transformed into *E. coli*, cloned, miniprepped and subjected to restriction digestion, as previously described. The engineered *Agrobacterium* culture was stored as glycerol stock at −80 °C.

A week before plant transformation, glycerol stocks of the engineered *A. tumefaciens* AGL1 strains harboring pCambia2300 RNAi vectors were plated for 3 days on YM medium containing 25 µg/mL of rifampicin, 250 µg/mL of carbenicillin and 50 µg/mL of kanamycin. Single colonies were used to inoculate 5 mL YM broth, supplemented with selection antibiotics prior to incubation at 28 °C at 200 rpm for 3 days. One day prior to transformation, 5 mL of the preculture was added to 20 mL of freshly prepared Luria Bertani (LB) broth [115,116] supplemented with selection antibiotics and incubated on an orbital shaker (150 rpm) at 28 °C until the OD_600nm_ reached 0.8. The bacterial cells were harvested by centrifugation at 5000× *g* for 10 min at 4 °C and resuspended in 25 mL of TMA1 medium (MS Macro, MS Micro, MS vitamins, 1 mg/L of biotin, 100 mg/L of malt extract, 100 mg/L of glutamine, 230 mg/L of proline, 40 mg/L of ascorbic acid, 5 g/L of PVP 10, 200 mg/L of cysteine, 1 mg/L of IAA, 1 mg/L of NAA, 4 mg/L of 2,4-D, 85.5 g/L of sucrose, pH 5.3) supplemented with 100 μM acetosyringone. The bacterial suspension was incubated at 28 °C for 3 h with shaking at 150 rpm. Towards the end of this culture phase, the optical density (OD_600nm_) of the culture was checked and adjusted to 0.6 with TMA1 medium.

#### 4.6.2. Transformation with Agrobacterium

Embryogenic cell suspensions (ECSs) from NARO, previously prepared from male flowers of East African Highland Banana cultivar Nakitembe and maintained on MA2 liquid media [117], were transformed using Centrifugation-Assisted *Agrobacterium* Transformation [114]. Briefly, fast cell division was initiated by transferring 1.25 mL settled cell volume (SCV) of ECSs to 50 mL of freshly prepared MA2,l which was maintained at 25 °C for five days at 100 rpm. On the 6th day, 0.5 mL of SCV was resuspended in 10 mL of an acetosyringone-activated *Agrobacterium* suspension, adjusted to 0.6 (OD_600nm_) [118,119]. The mixture of ECS and *Agrobacterium* was incubated in the dark at 22 °C for 5 days before washing in liquid MA2 supplemented with the broad-spectrum antibiotic cefotaxime. The clean cells were placed on meshes which were transferred onto selective embryo formation media (MA3), sub-cultured at biweekly intervals for 3 months, then transferred to RD1, micropropagation and rooting media, as previously described [120]. At least 50 lines were regenerated for further analysis for each construct (except for *Ego-1*, where the transformation procedure was stopped due to contamination).

#### 4.6.3. PCR Analysis of Putatively Transformed Banana Plantlets

PCR analysis of putative transgenic lines, obtained from independent transformation events, was performed to confirm transformation. Shoots were taken from in vitro plantlets. Genomic DNA was extracted from 1 g of leaf tissue using a modified CTAB method [121]. The DNA was subjected to PCR analysis to detect the presence of the *npt*II gene with a predicted fragment size of 646 bp. Plasmid pCAMBIA2300 RNAi constructs and gDNA from a nontrans genic plant were used as a positive and negative control, respectively.

Then, three primer combinations with similar melting temperatures were carefully designed for use in a multiplex PCR (Appendix A), to detect the selectable marker *npt*II, the loop region and the different target genes using a specific primer targeting the sense sequence, as illustrated in Figure 3. PCR amplification of gDNA was performed in a 25 µL reaction volume using Gotaq master mix (Promega, Madison, WI, USA) with approximately 250 ng of DNA as well as forward and reverse primers. The amplification comprised a denaturation step for 5 min at 95 °C; subsequently, 40 cycles consisting of 1 min at 95 °C, 1 min at 56 °C and 1 min at 72 °C were performed, followed by a 10 min final extension step at 72 °C and 4 °C storage step. The PCR products were analyzed on a 1.5% agarose gel and viewed after staining with ethidium bromide.

#### 4.6.4. Preparation of Plants for Infection Tests

Out of 50 PCR positive lines obtained, 30 vigorously growing lines per construct were selected along with untransformed control plantlets for micropropagation on proliferation medium (MS salts and vitamins, 10 mg/L of ascorbic acid, 100 mg/L of myo-inositol, 5 mg/L of BAP, 30 g/L of sucrose, 3 g/L of gelrite, pH 5.8) in order to generate at least 5 clones of each line. The individual shoots were then transferred to rooting medium (MS salts and vitamins, 10 mg/L of ascorbic acid, 100 mg/L of myo-inositol, 1 mg/L of IBA, 30 g/L of sucrose, 3 g/L of gelrite, pH 5.8). From the micropropagation, five shoot clones (replicates) were grown per transgenic line. Five well-rooted clones from each line (transgenic and control) were weaned in small disposable plastic cups (10 cm diameter) containing sterile soil, transferred to a transparent polythene chamber within a contained (Biosafety level II) greenhouse and they were grown for 4 weeks under diffused light, high humidity and at 26–28 °C. After 3 weeks, humidity was progressively reduced by gradual opening of the chamber. After 4 weeks, plants were transferred to 30 cm diameter pots in the greenhouse and irrigated manually on alternate days. Out of the five clones from each line, three were transferred to the screenhouse after two months and maintained for another month prior to infection with *R. similis*.

#### 4.6.5. Infection Procedure

*R. similis* isolated from roots of infected banana plants were multiplied on carrot discs and extracted as described in Section 4.2. A working stock of 400 nematodes per mL was made by dilution. Each plant was inoculated with 2000 nematodes in a 5 mL volume dispensed into falcon tubes. The pots were inoculated by depositing the suspension in a hole at the base of the shoot. The plants were allowed to grow for 3 months prior to assessment of nematode damage.

#### 4.6.6. Assessing Damage and Nematode Reproduction

At thirteen weeks post inoculation, growth parameters, height and girth were recorded and damage was assessed from each plant as previously reported [122], with root necrosis as the key parameter. Other parameters included fresh root weight, number of dead roots, live roots, nematode counts and plant height.

##### Assessing Root Damage

The plants were gently removed from the pots onto a polythene mat, where soil was loosened from the roots prior to rinsing with water. The clean roots were cut off from the corm, and excess water was soaked off using soft tissue. Plant height, pseudostem girth, and number of healthy leaves (where 70% of the leaf surface was healthy) were measured. Root damage was assessed by counting the number of dead and functional roots (live roots). To estimate necrosis index, 5 functional (live) root pieces were randomly selected and a 10 cm section split into 2 longitudinal sections. One half was used to score the percentage root necrosis by estimating the amount of damaged cortical tissue on a scale of 0–20 [123]. The split halves of the roots were placed into a labelled sample bag for use in nematode extraction. The rest of the root system was placed in a polythene bag, labelled and taken to the lab for nematode extraction. For soil extraction, about 1 Kg of soil from each pot was sampled into a labelled polythene bag.

##### Extraction of Nematodes from Roots

Nematodes were extracted from roots by maceration–filtration using a modification of the Baermann funnel technique (Hooper, 1986). The 5 root pieces previously used to assess percentage root necrosis, together with their other halves (totaling up to 10 longitudinal sections), were used for nematode extraction in a procedure that involved maceration, subsampling, filtration and decanting. Briefly, the roots were chopped with a kitchen knife on a chopping board to a length of approximately 1 cm. The samples were mixed and a subsample of 5 g was macerated in an electric blender for about 15 s (Hooper 1986). The macerated root suspension was poured onto sieve tissues (double-ply tissue paper, wet-strength), supported on plastic screens with a 2 mm mesh. After leaving the samples to stand for 48 h, the filter assembly was lifted out of the plates and the nematode containing extract was gently swirled and poured into beakers. After the nematodes in the suspension settled, the supernatant was decanted and the nematodes were transferred in a total of 25 mL to vials and kept at room temperature to establish counts.

##### Extraction of Nematodes from Soil

The sampled soil was mixed thoroughly, and a subsample of 100 g was taken for nematode extraction. This was then uniformly spread over a fine double-ply, wet-strength tissue supported on plastic screens with a mesh size of 2 mm. Water was added slowly down the sides of the plastic plate, just enough water to cover the soil. This was then incubated on the bench at room temperature for 24 h before the screens were lifted and the nematode suspension transferred to a beaker for decanting, as described in Extraction of Nematodes from Roots.

##### Characterization and Counting of Nematodes

Nematode characterization and counting were performed with a stereomicroscope (dissecting microscope). Aliquots of 5 mL were drawn from the 25 mL nematode suspension in the vials, put in a grid-marked counting dish and placed on a microscope stage, where it was left undisturbed for 5 min to allow the nematodes to settle at the bottom of the dish. Counts of females, males and juveniles were performed for both root samples and soil samples. However, no nematodes could be recovered from the soil samples.

##### Statistical Data Analysis

Collected experimental data for the necrosis index and the female nematode populations of seven different plant groups, consisting of five groups expressing endogenous dsRNA target genes (dsRps13, dsUnc-87, dsChs-2, dsEng1a, dsPat-10), one nonendogenous control (dsCs-Laccase-2) and one nontransformed control (NTC) were statistically analyzed. Per plant group, 30 lines each comprising three biological replicates were used during the analysis (n = 90). Statistical data analysis was conducted using RStudio version 4.2.2. First, normality and homoscedasticity (assumption of equal variances) were checked at a 5% significance level with a Shapiro–Wilk and a Levene test, respectively. However, the data for both parameters (necrosis index and female nematode populations) were not normally distributed, so the rank-based nonparametric Kruskal–Wallis test (α = 0.05) was used to verify whether there is a significant difference between the seven plant groups. Kruskal–Wallis *p*-values for both parameters were less than 0.001, indicating significant differences between the groups tested. Multiple nonparametric pairwise comparisons were achieved with Dunn’s method (α = 0.05) to reveal significant differences among the plant groups.

## Figures and Tables

**Figure 1 ijms-24-12126-f001:**
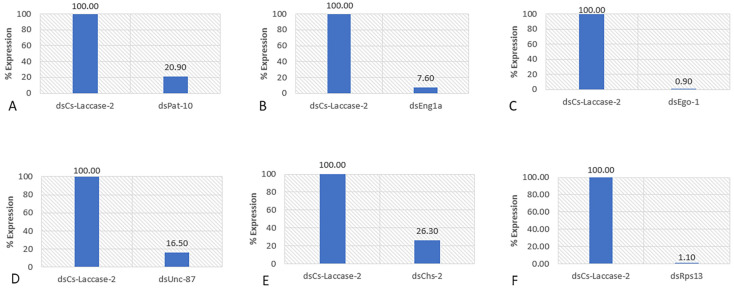
Relative gene expression of the target genes after a 36 h dsRNA soaking period, comparing the effect of dsRNA of the target gene to the nonendogenous dsRNA Cs-Laccase-2 control. Gene expression analyzed: (**A**) *Pat-10*; (**B**) *Eng1a*; (**C**) *Ego-1*; (**D**) *Unc-87*; (**E**) *Chs-2*; (**F**) *Rps13*. The dsRNA treatment is indicated under the bar. DsCs-Laccase-2 is the nonendogenous control. Two to three biological repeats were used for this analysis (see Section 4.5.5).

**Figure 2 ijms-24-12126-f002:**
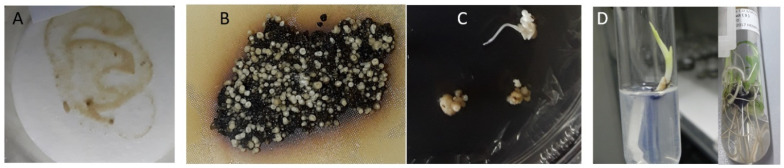
Selection and regeneration of putative transgenic plantlets. (**A**): Thinly spread embryogenic cell suspensions cocultivated with *Agrobacterium*; (**B**): Selection medium showing white embryos developing against a dark background of dead cells; (**C**): Well-developed somatic embryos germinating into shoots; (**D**): Plantlets on proliferation media.

**Figure 3 ijms-24-12126-f003:**
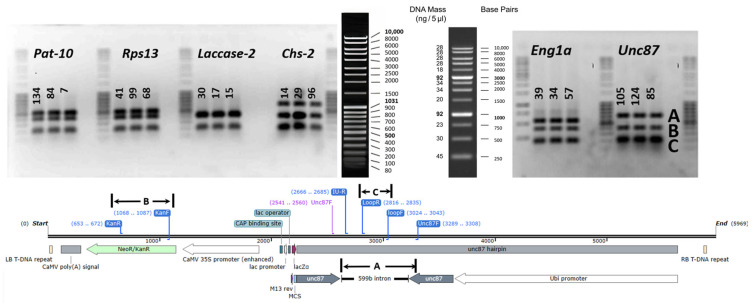
Location of the three primer pairs for detection of different T-DNA elements: kanamycin resistance gene (**B**), the hairpin loop (**C**) and a specific primer located in the target gene, shown here for *Unc-87* (**A**) with another primer located in the intron sequence. PCR gel images are shown for some of the lines (numbered) obtained with the constructs for the target genes indicated above the DNA bands.

**Figure 4 ijms-24-12126-f004:**
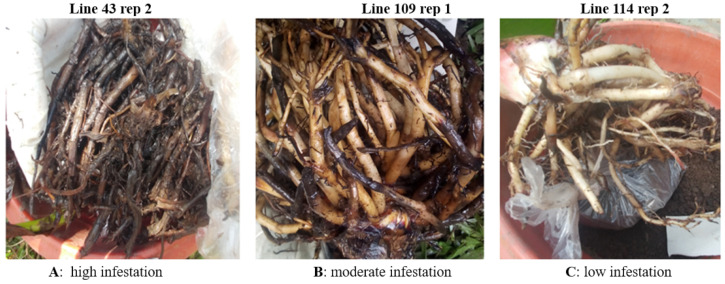
Representative examples of different root infestation levels in transformed lines expressing dsRps13 at three months after inoculation with 2000 *R. similis* nematodes.

**Figure 5 ijms-24-12126-f005:**
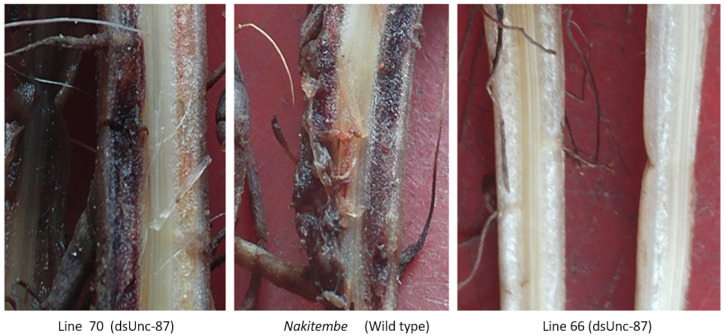
Some examples of the comparison of necrotic colorations in dissected roots three months after inoculation of each potted plant with 2000 *R. similis* nematodes.

**Figure 6 ijms-24-12126-f006:**
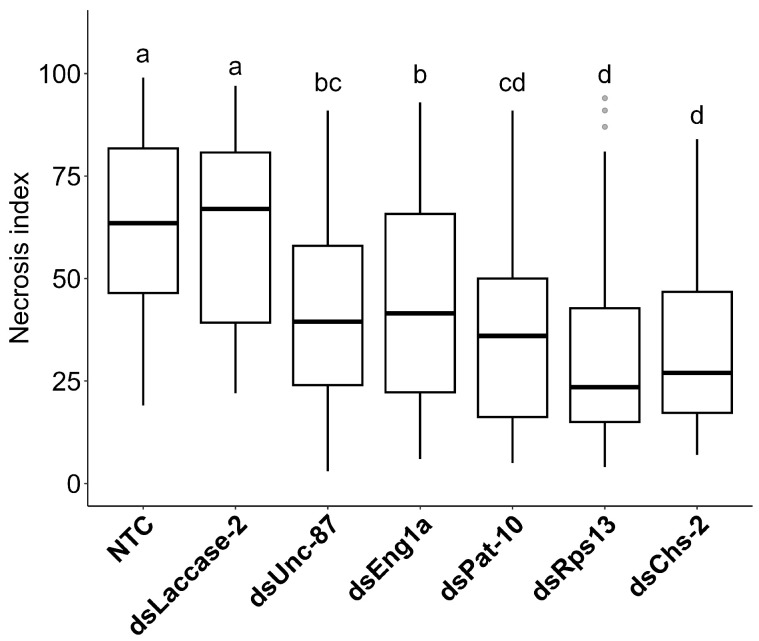
Box plot showing differences in necrosis between the various plant test groups. Medians denoted by the same letter are not significantly different (n = 90).

**Figure 7 ijms-24-12126-f007:**
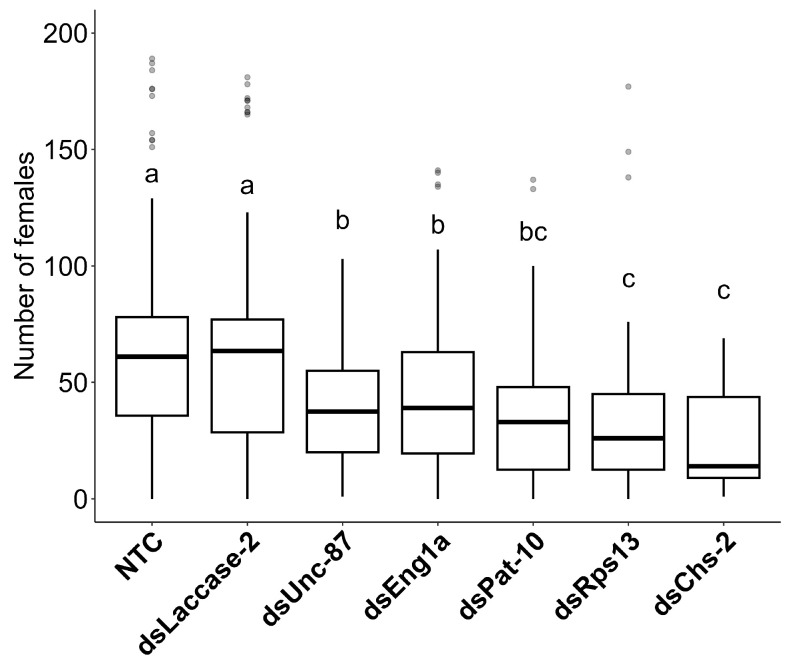
Box plot showing the nematode populations in the banana roots at three months after inoculation. The number is the female nematode count in 5 mL of the 25 mL extracted from 5 g of roots. Median counts denoted by the same letter are not significantly different.

**Table 1 ijms-24-12126-t001:** Mean nematode counts and standard deviation (SD) on 24 carrot discs for the different treatments.

dsRNA Target	Mean	Mean Separation	SD	Relative Population Counts
*Cs-Laccase-2*	14,285	a	1287	1
no dsRNA	13,757	a	1154	-
*Rs-Chs-2*	5170	b	530	0.36
*Rs-Pat-10*	3292	c	676	0.23
*Rs-Ego-1*	3021	c	477	0.21
*Rs-Unc-87*	2927	c	502	0.21
*Rs-Eng1a*	2858	c	559	0.20
*Rs-Rps13*	1833	d	464	0.13

**Table 2 ijms-24-12126-t002:** Lines offering the highest protection against nematodes based on lowest necrosis index scores.

dsRNA Target Genes	Best Lines	Protection (%)
*Rps13*	86, 127, 98, 92, 87, 93, 54, 142, 68	92–85
*Pat-10*	104, 80, 59, 84, 60, 103	92–85
*Unc-87*	66, 140, 139	92–85
*Chs-2*	24, 12, 44, 8	88–85
*Eng1a*	53, 58, 52	87–85

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
