# Peer review of "Transgenic East African Highland Banana Plants Are Protected against Radopholus similis through Host-Delivered RNAi"

_ijms, 2023, doi:10.3390/ijms241512126_

Round 1

Reviewer 1 Report

The authors have utilized an RNAi approach to enhance the resistance of bananas against nematodes. Through extensive in vitro and in vivo investigations, they have examined RNAi derived from various nematode genes, observing a remarkable effectiveness of the respective RNAi molecules in combating nematode infections. The study exhibits meticulous experimental design and a well-structured presentation.

However, one aspect that piques my curiosity is the choice of promoter. I believe that employing a root-specific promoter to drive the RNAi transgene would be more appropriate for nematode resistance, although it may not be an absolute requirement at this stage.

To summarize, this research holds substantial value for both researchers and breeders, providing promising insights not only into enhancing nematode resistance in bananas but also in other plant species.

Author Response

The authors have utilized an RNAi approach to enhance the resistance of bananas against nematodes. Through extensive in vitro and in vivo investigations, they have examined RNAi derived from various nematode genes, observing a remarkable effectiveness of the respective RNAi molecules in combating nematode infections. The study exhibits meticulous experimental design and a well-structured presentation.

However, one aspect that piques my curiosity is the choice of promoter. I believe that employing a root-specific promoter to drive the RNAi transgene would be more appropriate for nematode resistance, although it may not be an absolute requirement at this stage.

We completely agree with the reviewer that a root-specific promoter is the best approach for engineering nematode resistance in plants, at least in the case of the root parasites which are the most common. However, at the start of this project, there were no good banana root promoters known. Therefore, we have chosen to use a constitutive promoter. In 2022, James et al., reported some strong banana root promoters, this will allow to use these more specific promoters for the development of new constructs.

Reviewer 2 Report

The current study screened the potential target gene for controlling burrowing nematodes. The current study is interesting, and provides useful information. I only provide some suggestions for improving it.

(1)    Authors should introduce the background of selected genes.

(2)    Figure1: Any replicates for gene expression?

(3)    There should be more biological indicators for evaluating the control effects. For instance, disease index and control effect.

(4)    Can acid fuchsin mark the nematodes? Please provide the stained photos, and calculate the nematode number.

Author Response

(1).   Authors should introduce the background of selected genes.

The background of the genes was described in the discussion of the paper (see text below). We have moved it to the introduction to make it more visible to the reader. Is this sufficient or does the reviewer want a more detailed description for each gene? We have added some more details in the lower paragraphs for the reviewers and have incorporated the more recent references also in the new introduction.

This is the new text in the introduction:

The gene targets (Chs-2, Ego-1, Eng1a, Pat-10, Rps13 and Unc-87) for this study were selected based on promising reports from nematodes where adverse phenotypes were attributed to the dsRNA administered either through soaking or feeding. Chitin synthase (Chs-2) was selected because viable egg production requires chitin which is a key component of the eggshell (Clarke, Cox, & Shepherd, 1967). Ingestion of dsRNA by C. elegans and soaking eggs of Meloidogyne artiellia in Chs-2 dsRNA affected egg development (Fanelli, Di Vito, Jones, & De Giorgi, 2005).  Host induced gene silencing of chitin synthase conferred resistance to soybean cyst nematode in soybean (Kong et al., 2022). The Ego-1 gene encodes an RNA-directed RNA polymerase that is important in C. elegans germline development (Smardon et al., 2000). Endo-β-1,4-glucanase (Eng1a) hydrolyses cellulose allowing nematodes to degrade and penetrate cell walls (Haegeman, Jacob, Vanholme, Kyndt, & Gheysen, 2008) and RNAi has resulted in reduced nematode infection (Peng et al., 2014). The Unc-87 gene encodes actin-binding proteins essential for maintenance of the nematode body wall muscle which plays a critical role in worm motility (Kranewitter, Ylanne, & Gimona, 2001; Ono, Obinata, Yamashiro, Liu, & Ono, 2015). Related to movement is Pat-10 encoding a body wall muscle troponin C, essential for muscle contraction and successful embryonic morphogenesis (Kagawa, Takuwa, & Sakube, 1997). RNAi of Pat-10 has been shown to result in paralysis (‘walking stick’ phenotype), larval and embryonic lethality and maternal sterility in the nematode (Joseph, Gheysen, & Subramaniam, 2012; Nsengimana, Bauters, Haegeman, & Gheysen, 2013). The Rps13 gene encodes the ribosomal protein S13, a component of the 40S subunit of ribosomes that catalyze protein synthesis (Cukras et al., 2003). Rps13 RNAi in C. elegans resulted in severe embryo defects (Sonnichsen et al., 2005).

Additional information

Chitin synthase-2 (CHS-2) is a key enzyme involved in the critical process of chitin synthesis, which plays a vital role in nematodes (Zhang, Foster et al., 2005). Chitin, a structural polysaccharide, provides strength and protection to essential structures such as the nematode's pharynx, stylet and eggshell. Chs-1 is responsible for chitin synthesis in eggshells, which provide support and protection during embryonic development. Chs-2 exhibits expression in various tissues and developmental stages of nematodes, and also in the egg sac of plant parasitic nematodes (Fanelli, Di Vito et al., 2005). The disturbance of chitin synthesis or hydrolysis led to embryonic lethality, defective egg laying or molting failure (Chen & Peng, 2019). 

Ego-1 (Enhancer of glp-1) is an essential gene found in nematodes. It plays a crucial role in germ-line development and RNA interference pathways, which are vital for gene regulation and defense against transposable elements and viruses (Smardon et al., 2000). The importance of ego-1 in nematodes lies in its involvement in germ-line development. Studies have shown that putative ego-1 null mutants display multiple defects in germ-line development, highlighting its essential role in this process and potential for use in nematode control (Smardon et al., 2000).

The gene Eng1a encodes an endo-β-1,4-glucanase, which plays a crucial role in the degradation of cellulose, a complex carbohydrate found in plant cell walls. In nematodes, ENG-1A is of significant importance for breaking down the plant cell wall to allow them to migrate into and through the plant roots. RNAi-mediated knockdown of Eng1a in the plant-parasitic nematode Meloidogyne javanica or Heterodera glycines resulted in reduced reproduction .

Proline-rich and alanine-rich transmembrane protein 10 (PAT-10) has been associated with embryonic development, morphogenesis, locomotion, and neuronal processes in different nematode species. It is localized to neuronal structures and can also be detected in muscle cells, the pharynx, and other tissues. RNAi-mediated knockdown of Pat-10 in Caenorhabditis elegans resulted in impaired locomotion and altered neuronal processes (Kagawa et al., 1997).

The gene Rps-13 encodes the ribosomal protein S13 (RPS-13), an indispensable component of the ribosome, the cellular machinery responsible for protein synthesis. Its primary role involves mRNA decoding and facilitating the assembly of amino acids into polypeptide chains during translation. RNA interference (RNAi)-mediated silencing of Rps-13 has proven invaluable in uncovering the functional implications of RPS-13, including developmental defects, impaired reproduction, and reduced viability (Wang et al., 2020).

The gene Unc-87 (uncoordinated) encodes an actin-bundling protein (Kranewitter et al., 2001), contributing to the proper organization of muscle filaments, facilitating the assembly and alignment of myosin and actin filaments within the sarcomere. Mutations of Unc-87 in C. elegans cause disorganization of sarcomeric actin filaments in body wall muscle, but this phenotype is suppressed when muscle contraction is reduced, suggesting that UNC-87 stabilizes actin filaments during actomyosin contraction (Goetinck & Waterston, 1994). Mutations or disruptions in Unc-87 can lead to abnormal muscle structure, impaired movement, and locomotor defects in nematodes (Yamashiro, Gimona, & Ono, 2007).

(2)    Figure 1: Any replicates for gene expression?

Six biological replicates were made for each dsRNA treatment, namely dsChs-2, dsEgo-1, dsEng1a, dsRps13, dsPat-10, and dsUnc-87. These replicates were prepared in soaking tubes containing the nematode suspension and the soaking solution. The tubes were then incubated at 25°C for 36 hours on a rotary shaker set at 50 rpm. cDNA was extracted and after checking the quality, every two cDNA samples were combined to make three replicates for the RT-PCR.

We have added this information to the methods and the legend.

(3)    There should be more biological indicators for evaluating the control effects. For instance, disease index and control effect. 

Thank you for your feedback regarding the inclusion of a disease index along with the necrosis index and other data points to evaluate the control effects in our study. We appreciate your suggestion and understand the importance of incorporating such an approach for a more comprehensive assessment. While there is currently no published methodology specifically addressing the calculation of a disease index using the combination of different parameters in the context of nematode-infested plants, we recognize the need for a quantitative measure that integrates multiple biological indicators to evaluate the overall disease severity and control effects. In light of your suggestion, a suitable methodology could be developed to calculate the disease index based on the available data, by carefully analyzing the relationship between the necrosis index, number of dead roots, girth measurements, root fresh weight, Radopholus similis population dynamics, and plant height at different stages. By considering these factors collectively, a robust disease index could be derived that can effectively capture the impact of nematode infestation and evaluate the efficacy of control measures. Although the development of such a methodology may require additional experimentation and statistical analysis, we believe that it could be valuable to enhance the accuracy and comprehensiveness of our study. In case the reviewer finds this essential to do, we would need extra time to perform this analysis. On the other hand, it could be better to combine this extra information in a different, methodological paper. In any case, we believe that the ultimate proof of disease control lies in field experiments where the yield of the different plants is analyzed and correlated with the nematode population in the roots.

(4)   Can acid fuchsin mark the nematodes? Please provide the stained photos and calculate the nematode number.

Thank you for your inquiry regarding the use of acid fuchsin to mark nematodes in our study. In our experiment, we did not utilize acid fuchsin staining to mark the nematodes. While acid fuchsin staining can be a useful tool for differentiating and counting nematodes, we opted for alternative methods to assess and quantify the nematode populations because of the huge number of samples and the thickness of the roots of the screenhouse grown banana plants. However, we understand your desire for visual evidence and accurate quantification. To address your concern about the quantification, we provided detailed descriptions of the nematode extraction and counting procedures we employed in our study as outlined in sections 4.6.6.2-4.6.6.4 of the manuscript. We believe that these methods that are routinely used in the analysis of crop infection are adequate for a reliable quantification.

To extract nematodes from the roots, we employed a modified version of the Baermann funnel technique, following the maceration-filtration method described by Hooper (1986). Specifically, five root pieces, previously used to assess the percentage of root necrosis, were paired with their other halves, resulting in a total of up to ten longitudinal sections. These sections were chopped to approximately 1cm in length using a kitchen knife. A subsample of 5g from the chopped roots was macerated in an electric blender for approximately 15 seconds. The macerated root suspension was then poured onto double ply tissue paper (wet-strength) supported on plastic screens with a 2mm mesh. After a 48-hour incubation period, the filter assembly was lifted, and the nematode-containing extract was gently swirled and poured into beakers. Following settlement of nematodes in the suspension, the supernatant was decanted, and the nematodes were transferred to vials, totaling 25mL, and kept at room temperature for counting (Section 4.6.6.2). For nematode extraction from the soil, a thoroughly mixed subsample of 100g was uniformly spread over fine double ply tissue paper supported on plastic screens with a mesh size of 2mm. Water was added slowly down the sides of the plastic plate until the soil was just covered. This soil-water mixture was incubated at room temperature on a bench for 24 hours. Subsequently, the screens were lifted, and the nematode suspension was transferred to a beaker for decanting, following the procedure described in Section 4.6.6.2 (Section 4.6.6.3). Nematode characterization and counting were performed using a stereomicroscope (dissecting microscope). Aliquots of 5mL were drawn from the 25mL nematode suspension in the vials and placed in a grid-marked counting dish on a microscope stage. The dish was left undisturbed for 5 minutes to allow the nematodes to settle at the bottom. Counts of females, males, and juveniles were conducted for both root and soil samples. However, it should be noted that no nematodes were recovered from the soil samples (Section 4.6.6.4). Although staining the nematodes with acid fuchsin was not part of the current study, we appreciate your suggestion and acknowledge the potential benefits it offers for future investigations. We will consider incorporating acid fuchsin staining in subsequent studies to enhance visualization and quantification of nematodes. However, this will require sectioning of the thicker roots and therefore it cannot be used for reliable quantification. Thank you for your valuable input and your understanding of the limitations in our experiment. We hope that the provided information on the nematode extraction and counting methods, as described in sections 4.6.6.2-4.6.6.4, addresses your inquiry adequately.

James A, Paul J-Y, Souvan J, Cooper T, Dale J, Harding R and Deo P (2022) Assessment of root specific

promoters in banana and tobacco and identification of a banana TIP2 promoter with strong root activity. Front. Plant Sci. 13:1009487. doi: 10.3389/fpls.2022.1009487

Chen, Q., & Peng, D. (2019). Nematode Chitin and Application. Adv Exp Med Biol, 1142, 209-219. doi:10.1007/978-981-13-7318-3_10

Clarke, A., Cox, P., & Shepherd, A. (1967). The chemical composition of the egg shells of the potato cyst-nematode, Heterodera rostochiensis Woll. Biochemical Journal, 104(3), 1056-1060. doi:10.1042/bj1041056

Cukras A.R., Southworth D.R., Brunelle J.L., Culver G.M., Green R. (2003). Ribosomal proteins S12 and S13 function as control elements for translocation of the mRNA:tRNA complex. Mol Cell. 2003 Aug; 12(2):321-8. doi: 10.1016/s1097-2765(03)00275-2. PMID: 14536072.

Fanelli, E., Di Vito, M., Jones, J. T., & De Giorgi, C. (2005). Analysis of chitin synthase function in a plant parasitic nematode, Meloidogyne artiellia, using RNAi. Gene, 349, 87-95. doi:10.1016/j.gene.2004.11.045

Goetinck, S., & Waterston, R. H. (1994). The Caenorhabditis elegans muscle-affecting gene unc-87 encodes a novel thin filament-associated protein. The Journal of cell biology, 127(1), 79-93.

Haegeman, A., Jacob, J., Vanholme, B., Kyndt, T., & Gheysen, G. (2008). A family of GHF5 endo-1,4-beta-glucanases in the migratory plant-parasitic nematode Radopholus similis. Plant Pathology, 57(3), 581-590. doi:https://doi.org/10.1111/j.1365-3059.2007.01814.x

Hooper, D.J. (1986). Handling, fixing, staining and mounting nematodes. In: Southey, J.F. (Ed.). Laboratory Methods for Work with Plant and Soil Nematodes. London, UK: CAB International, pp. 59–80.

Hu L, Cui R, Sun L, Lin B, Zhuo K, Liao J. Molecular and biochemical characterization of the β-1,4-endoglucanase gene Mj-eng-3 in the root-knot nematode Meloidogyne javanica. Exp Parasitol. 2013 Sep;135(1):15-23. 

Joseph, S., Gheysen, G., & Subramaniam, K. (2012). RNA interference in Pratylenchus coffeae: knock down of Pc-pat-10 and Pc-unc-87 impedes migration. Mol Biochem Parasitol, 186(1), 51-59. doi:10.1016/j.molbiopara.2012.09.009

Kagawa, H., Takuwa, K., & Sakube, Y. (1997). Mutations and expressions of the tropomyosin gene and the troponin C gene of Caenorhabditis elegans. Cell structure and function, 22(1), 213-218.

Kong, L., Shi, X., Chen, D., Yang, N., Yin, C., Yang, J., . . . Liu, S. (2022). Host-induced silencing of a nematode chitin synthase gene enhances resistance of soybeans to both pathogenic Heterodera glycines and Fusarium oxysporum. Plant Biotechnology Journal, 20(5), 809-811. doi:https://doi.org/10.1111/pbi.13808

Kranewitter, W. J., Ylanne, J., & Gimona, M. (2001). UNC-87 Is an Actin-bundling Protein *. Journal of Biological Chemistry, 276(9), 6306-6312. doi:10.1074/jbc.M009561200

Mb, B., Urwin, P. E., & Atkinson, H. J. (2007). qPCR Analysis and RNAi Define Pharyngeal Gland Cell-Expressed Genes of Heterodera glycines Required for Initial Interactions with the Host. Molecular plant-microbe interactions, 20, 306-312. doi:10.1094/MPMI-20-3-0306

Nsengimana, J., Bauters, L., Haegeman, A., & Gheysen, G. (2013). Silencing of Mg-pat-10 and Mg-unc-87 in the plant parasitic nematode Meloidogyne graminicola using siRNAs. Agriculture, 3(3), 567-578.

Ono, K., Obinata, T., Yamashiro, S., Liu, Z., & Ono, S. (2015). UNC-87 isoforms, Caenorhabditis elegans calponin-related proteins, interact with both actin and myosin and regulate actomyosin contractility. Molecular biology of the cell, 26(9), 1687-1698.

Peng, H., Peng, D., Long, H., He, W., Qiao, F., Wang, G., & Huang, W. (2014). Characterisation and functional importance of β-1,4-endoglucanases from the potato rot nematode, Ditylenchus destructor. Nematology, 16(5), 505-517. doi:https://doi.org/10.1163/15685411-00002783

Smardon, A., Spoerke, J. M., Stacey, S. C., Klein, M. E., Mackin, N., & Maine, E. M. (2000). EGO-1 is related to RNA-directed RNA polymerase and functions in germ-line development and RNA interference in C. elegans. Curr Biol, 10(4), 169-178. doi:10.1016/s0960-9822(00)00323-7

Sonnichsen B., Koski L.B., Walsh A., Marschall P., Neumann B., Brehm M., Alleaume A.M., Artelt J., Bettencourt P., Cassin E., Hewitson M., Holz C., Khan M., Lazik S., Martin C., Nitzsche B., Ruer M., Stamford J., Winzi M., Heinkel R., Roder M., Finell J., Hantsch H., Jones S.J., Jones M., Piano F., Gunsalus K.C., Oegema K., Gonczy P., . . . Echeverri C.J. (2005). Full-genome RNAi profiling of early embryogenesis in Caenorhabditis elegans. Nature, 434, 462-9. doi:10.1038/nature03353

Wang, M., Chen, X., Wu, Y., Zheng, Q., Chen, W., Yan, Y., . . . Zheng, B. (2020). RpS13 controls the homeostasis of germline stem cell niche through Rho1‐mediated signals in the Drosophila testis. Cell proliferation, 53(10), e12899.

Yamashiro, S., Gimona, M., & Ono, S. (2007). UNC-87, a calponin-related protein in C. elegans, antagonizes ADF/cofilin-mediated actin filament dynamics. Journal of cell science, 120, 3022-3033. doi:10.1242/jcs.013516

 Zhang, Y., Foster, J. M., Nelson, L. S., Ma, D., & Carlow, C. K. (2005). The chitin synthase genes chs-1 and chs-2 are essential for C. elegans development and responsible for chitin deposition in the eggshell and pharynx, respectively. Developmental biology, 285(2), 330-339.

Reviewer 3 Report

The manuscript is well-written and well-designed. There are some minor questions that could be addressed and added to the manuscript.

For Figure1, how many replicates were used for the imageJ analysis. The authors should mention this in the methods. Also, there are no standard errors in the bars.

Figure 3. Did the authors test the production of hairpin RNA in the plants? Or did they just do a PCR for the transgene? It is essential to check the expression of the hairpin RNA for effective RNAi.

Author Response

Figure 1.

Thank you for your valuable feedback. We apologize for the oversight in not mentioning the number of replicates used for the ImageJ analysis in the methods section. To address your concern, we would like to clarify that two replicates were used for the Pat-10 gene analysis, while three replicates were employed for the analysis of the other genes. This information has been added to the methods section to provide a comprehensive description of the experimental design.

Six biological replicates were made for each dsRNA treatment, namely dsChs-2, dsEgo-1, dsEng1a, dsRps13, dsPat-10, and dsUnc-87. These replicates were prepared in soaking tubes containing the nematode suspension and the soaking solution. The tubes were then incubated at 25°C for 36 hours on a rotary shaker set at 50 rpm. cDNA was extracted and after checking the quality, every two cDNA samples were combined to make three replicates for the RT-PCR.

Regarding the absence of standard errors in the bars, we want to explain that the nature of our analysis and the computed gene expression values did not permit the inclusion of error bars. The quantification of gene expression was based on the relative intensity of the PCR products, as analyzed by ImageJ. Our aim was to compare the expression levels of the treatment genes with the internal actin control using a relative measurement approach. Following the pooling of cDNA samples, RT-PCR was performed on each pooled cDNA sample. This means that for pat-10 treated samples, we performed RT-PCR on 2 pooled cDNA samples, and for the other genes with 3 biological replicates, we performed RT-PCR on 3 pooled cDNA samples.  After the RT-PCR analysis, gel electrophoresis was conducted, and the resulting images were analyzed using ImageJ software. The data obtained from the gel electrophoresis analysis allowed us to compute an average expression level for each of the genes normalized to the actin control average and no error bars could be generated with this method. Nevertheless, to take care of the reviewer’s concern, and provide insights into the variability and potential error in our data, an alternative visualization is provided here. Individual plots were generated, capturing the expression levels of the gene in each sample separately, the first three (or two for pat-10) are control treatment (nematodes soaked in non-endogenous dsRNA dslaccase-2), the others are from nematodes soaked in the target ds treatment. By utilizing both the average expression (Figure 1) and individual plots, we are able to obtain a more comprehensive understanding of the gene expression profiles.

Figure 3. Did the authors test the production of hairpin RNA in the plants? Or did they just do a PCR for the transgene? It is essential to check the expression of the hairpin RNA for effective RNAi.

The transgenic plants have been tested for the presence of the specific hairpin construct and for the expression of the selectable marker gene, to be sure that the T-DNA was inserted in euchromatin. Testing the expression level of the dsRNA is not very reliable, as the dsRNA should be degraded rapidly because of the RNAi. We agree that it would be good to check the level of the target gene expression in the nematodes recovered from the transgenic plants, but as the best performing plants had no or very few nematodes, this was not possible. Furthermore, in the current experiment, the number of tested plants was too large to include a molecular analysis. In the future, a selection of plants with different levels of nematode reproduction (very high, high, intermediate, low, very low) could be tested again including a molecular analysis to see if any correlation can be found between the level of resistance and either dsRNA or siRNA in the plant or target gene expression in the nematodes that can be recovered from the roots.

Round 2

Reviewer 2 Report

I have no further comments on it